# Antigen-Dependent Adjuvanticity of Poly(lactic-co-glycolic acid)-polyethylene Glycol 25% Nanoparticles for Enhanced Vaccine Efficacy

**DOI:** 10.3390/vaccines13030317

**Published:** 2025-03-16

**Authors:** Minxuan Cui, Jiayue Xi, Zhuoyue Shi, Yupu Zhu, Zhengjun Ma, Muqiong Li, Qian Yang, Chaojun Song, Li Fan

**Affiliations:** 1Shaanxi Key Laboratory of Chiral Drug and Vaccine Adjuvants, Department of Pharmaceutical Chemistry and Analysis, School of Pharmacy, Air Force Medical University, 169th Changle West Road, Xi’an 710032, China; cuiminxuan2521@fmmu.edu.cn (M.C.); 0518xjy@fmmu.edu.cn (J.X.); shizhuoyue2021@163.com (Z.S.); mazhengjun@fmmu.edu.cn (Z.M.); limuqiong1981@fmmu.edu.cn (M.L.); 2Department of Occupational & Environmental Health, The Ministry of Education Key Lab of Hazard Assessment and Control in Special Operational Environment, School of Public Health, Air Force Medical University, 169th Changle West Road, Xi’an 710032, China; zhuyupupapa@fmmu.edu.cn; 3Department of Chinese Materia Medical and Natural Medicines, School of Pharmacy, Air Force Medical University, 169th Changle West Road, Xi’an 710032, China; yangqian@fmmu.edu.cn; 4School of Life Science, Northwestern Polytechnical University, 127th Youyi West Road, Xi’an 710072, China

**Keywords:** antigen, nanovaccine, *Staphylococcus aureus*, nanoparticle adjuvant, adjuvanticity, PLGA, subunit vaccine

## Abstract

**Background**: A key component in modern vaccine development is the adjuvant, which enhances and/or modulates the antigen-specific immune response. In recent years, nanoparticle (NP)-based adjuvants have attracted much research attention owing to their ability to enhance vaccine potency. Nonetheless, how the selection of different antigens influences the overall vaccine efficacy when combined with the same nanoparticle adjuvant is less discussed, which is important for practical applications. **Methods**: Non-toxic mutants of exotoxin Hla (rHlaH35L) and cell-wall-anchored protein SpA(rSpam) were covalently conjugated to Poly(lactic-co-glycolic acid)-polyethylene glycol (PLGA-PEG) 25% NPs (25% NPs) as antigens to prepare nanovaccines. Antibody titers, cytokine secretion levels, and the antibody bacteriolytic capacity were tested to investigate immune activation. To evaluate the protective efficacy of the nanovaccine, immunized mice were challenged with *S. aureus* ATCC 25923 at three different lethal doses: 1 × LD_100_, 2 × LD_100_, and 4 × LD_100_. **Results**: We showed that 25% NP-rHlaH35L nanovaccines were associated with more efficient humoral, cellular, and innate immune responses and protection potency compared with 25% NP-rSpam. Moreover, the overall vaccine potency of 25% NP-rHlaH35L was even better than the combination vaccination of both 25% NP-rHlaH35L and 25% NP-rSpam. In comparison to the clinically used aluminum (alum) adjuvant, the 25% NP adjuvants were found to stimulate humoral and cellular immune responses efficiently, irrespective of the antigen type. For antigens, either exotoxins or cell-wall-anchored proteins, the 25% NP-based vaccines show excellent protection for mice from *S. aureus* infection with survival rates of 100% after lethal challenge, which is significantly superior to the clinically used alum adjuvant. Moreover, due to the superior immune response elicited by 25% NP-rHlaH35L, the animals inoculated with this formulation survived even after two times the lethal dose of *S. aureus* administration. **Conclusions**: We demonstrated that the type of antigen plays a key role in determining the overall vaccine efficacy in the immune system when different kinds of antigens are conjugated with a specific nanoparticle adjuvant, paving a new way for vaccine design based on 25% NP adjuvants with enhanced potency and reduced side effects.

## 1. Introduction

*Staphylococcus aureus* (*S. aureus*) is a Gram-positive commensal bacterium that can cause severe opportunistic infections, such as bacteremia, sepsis, and endocarditis [1]. In recent years, *S. aureus* has become a public health concern due to the emergence and spread of methicillin-resistant *S. aureus* (MRSA) strains that are resistant to most antibiotic treatments [2]. Vaccination has been envisioned to be an effective solution [3], but the development of *S. aureus* vaccines via the conventional approach of generating antibodies against surface antigens has appeared to be insufficient [4], due to immunoevasive adaptations and the lack of cellular response activation [5]. Extensive research supports the notion that T-cell-mediated immunity is a pivotal factor in *S. aureus* vaccine efficacy. For instance, while B-cell-deficient mice show no increased vulnerability to systemic *S. aureus* infection compared to wild-type mice [6], mice deficient in T cells [7], IFN-γ [8,9], TNF [10], or dual IL-17A/F [11] demonstrate heightened susceptibility to such infections. This evidence highlights the necessity for novel vaccine designs and approaches to achieve protection against *S. aureus*. At its core, a successful vaccine must integrate pathogen-specific antigens with immunostimulatory adjuvants to trigger a protective adaptive immune response. An effective immune response against *S. aureus* necessitates a multifaceted approach, involving antibodies that directly neutralize bacterial viability and toxicity, enhance opsonophagocytic activity, and promote T-cell-mediated immunity to recruit phagocytic cells to the site of infection [12].

The selection of *S. aureus*-specific antigens and adjuvants for immunostimulation are two critical components in developing a protective *S. aureus* vaccine. Antigens determine the targeting of the immune protection of vaccines and the protective spectrum of vaccines. A principle for selecting an antigen should be its conservation among different strains. Exotoxins and cell-wall-anchored (CWA) proteins are usually selected as antigens, which are widely expressed in most *S. aureus* strains for the following reasons: (1) Functionally, the antigen modified by exotoxins can block the process of inflammation and target cell lysis [13] and the antigen using CWA proteins can block bacterial adhesion colonization, nutrient acquisition, and immune evasion [14]. (2) In terms of secretion and expression characteristics, exotoxins and CWA proteins can simulate the process from bacterial adhesion to virulence, which has the advantage of biomimicry.

On the other hand, the adjuvant determines the production time, titer, and maintenance time of neutralizing antibodies, which influence the protective efficacy of the bacterial vaccine [15,16]. In fact, in animal models, antigen-specific antibody titers, antigen-specific T cell responses, and bacterial clearance during infection all increased with the addition of adjuvants [17]. In recent years, nanoparticles have been widely used in the research of new vaccine adjuvants due to their ease of modification and high efficiency [18]. Adjusting the size and shape of nanoparticles can alter the type and degree of vaccine immune activation [19]. Spherical polystyrene nanoparticles with a size of 193 nm produced a TH1-biased immune response, while larger rod-shaped polystyrene particles (1530 nm) induce a stronger TH2-biased immune response [20]. The rigidity of the nanoparticles also appears to change their pharmaceutical response characteristics, and it has been reported that the doping ratio of polyethylene glycol(PEG) in poly(lactic-co-glycolic acid)(PLGA) nanoparticles modulates the rigidity of PLGA nanoparticles. The rigidity of PLGA nanoparticle adjuvants affects the phagocytic ability of dendritic cells (DCs) and the rate of antigen release, ultimately leading to differences in vaccine protection. In short, a nanovaccine prepared with 25% PEG-doped PLGA nanoparticles combined with the *S. aureus* antigen rEsxB provided 100% protection in a mouse *S. aureus* infection model during subcutaneous immunization [21]. Wang et al. prepared negatively charged core-shell nanomicelles with different rigid cores using PLGA and PCL with varying proportions of LA and GA as soft cores and PSt as hard cores. They investigated the effect of core rigidity on the uptake efficiency of tumor cells. The results showed that the rigidity of the nanomicelle core significantly affected the uptake efficiency of the tumor cells. PLGA (LA/GA, 48/48, mole ratio) nanoparticles with a soft core structure exhibited the highest uptake efficiency by tumor cells [22]. Gao et al. prepared core–shell PLGA–liposome nanoparticles with different rigidities and studied their transport ability in tissues covered with biohydrogel. The results showed that the moderately rigid nanoparticles are deformed into ellipsoids, which promotes the rapid diffusion of nanoparticles in the hydrogel mesh. The encapsulation of doxorubicin by moderately rigid nanoparticles significantly increased the bioavailability of the drug in vivo [23].

Moreover, adjuvants have a clear selective preference for specific antigens [24]. For example, polyvinylimine and chitosan can enhance the binding level of antigens to mucosal surfaces for pathogens that invade through mucosal pathways [25]. The complex life cycle of *Plasmodium* and the variety of antigens put forward higher requirements for the development of malaria vaccines [26], and accordingly, multi-functional adjuvants are needed. The RTS,S vaccine is a malaria vaccine candidate that targets the spore phase of *Plasmodium* infection [27]. AS01 was selected as the vaccine adjuvant, consisting of 3-O-deacyl-4′-monophospholipid A (MPL) and saponin QS-21. MPL is a toll-like receptor-4(TLR4) agonist that induces mouse T cells to produce IFN-γ and convert antibodies to IgG 2a/c, while QS-21 induces the production of neutralizing antibodies and cytotoxic T cell responses [28]. Animal studies have shown that the RTS,S vaccine induces a rapid, instantaneous innate immune response in the injection site, draining lymph nodes, and produces antibody titers 20 times higher than natural exposure [29]. These studies suggested that adjuvants may also have a selective preference for different types of *S. aureus* antigens.

Herein, as illustrated in Figure 1, in order to mimic *S. aureus*, one exotoxin (Hla) and one CWA protein (SpA) were selected as vaccine antigens. PLGA nanoparticles with a 25% PEG doping ratio were chosen as the adjuvant. rHlaH35L and rSpam, the non-toxic mutants of Hla and the CWA protein SpA of *S. aureus*, were first prepared as recombinant antigens. Covalent bonding was also chosen as the conjugation method between the antigen and the adjuvant. The immunoactivation efficacy, protection, and biosafety issues have been well-studied. Moreover, the antigen selection preference of the 25% PLGA-PEG nanoparticles (25% NPs) adjuvant was also tested. We aimed to obtain an optimized nanoparticle vaccine formulation with the most protective effect to pave the way for future clinical trials.

## 2. Materials and Methods

### 2.1. Expression and Purification of rHlaH35L and rSpam Antigen

α-hemolysin (Hla), a critical exotoxin produced by *S. aureus*, plays a significant role in the pathogen’s infection process. Hla binds to the ADAM-10 receptor on target cell membranes, leading to cell lysis. To eliminate its hemolytic toxicity, the histidine at position 35 in the *hla* sequence was substituted with leucine, resulting in the HlaH35L plasmid [30,31,32]. Protein A (SpA), a cell-wall-anchored protein expressed by *S. aureus*, is essential for immune evasion. The immunoglobulin-binding domains (E, D, A, B, and C) of SpA are frequently employed as vaccine antigens. In wild-type SpA, two consecutive glutamines (QQ) at positions 7–8 and two consecutive aspartic acids (DD) at positions 34–35 within its five structural domains are crucial for immune escape. To generate a SpA variant, glutamines at positions 7–8 in each domain were replaced with lysine, and aspartic acids at positions 35–36 were substituted with alanine [33,34]. The expression plasmid was transformed into BL21 (DE3) *E. coli* (Tsingke Biotechnology Co., Ltd, Beijing, China), and recombinant proteins rHlaH35L and rSpam, both carrying a 6 × His-tag, were produced using standard IPTG induction [35]. These proteins were purified from the lysate of IPTG-induced BL21 (DE3) *E. coli* using Ni-Sepharose column chromatography. The Ni-Sepharose column (QIAGEN, Shanghai, China) was washed with phosphate buffer (PBS) and eluted with 100 mmol/L, 300 mmol/L, and 500 mmol/L imidazole gradients(Merck, Shanghai, China). The purified proteins were dialyzed into PBS buffer, and endotoxins were removed using a High-Capacity Endotoxin Removal column (Xiamen Bioendo Technology Co., Ltd., Xiamen, China). Protein concentrations were quantified using the BCA assay(Thermo Fisher Scientific, Waltham, MA, USA), and purity was assessed by SDS-PAGE.

### 2.2. Preparation and General Characterizations of 25% NPs Conjugated with Antigen

The preparation of PLGA-PEG 25% nanoparticles (25% NPs) followed a precipitation/solvent diffusion method, adapted from our previously established procedures with slight modifications [21]. In detail, 100 mg of PLGA_15K_-PEG_5K_-COOH polymers (50:50 lactic acid to glycolic acid ratio, Xi’an Ruixi Biological Technology Co., Ltd., Xi’an, China) were dissolved in an organic solvent mixture containing 1.5 mL of dichloromethane and 1 mL of acetone. This organic solution was slowly added to 10 mL of a 5% aqueous PVA solution (Shanghai Macklin Biochemical Technology Co., Ltd., Shanghai, China) under constant stirring at 600 rpm. The mixture was emulsified using an ultrasonic probe to form an oil-in-water emulsion. The emulsion was then transferred into 50 mL of DI water and stirred overnight. The 25% NP suspension was filtered through a 0.45 μm (Merck, Shanghai, China) filter to remove aggregates and centrifuged at 18,000× *g* at 4 °C for 1 h. The resulting precipitate was resuspended in DI water and centrifuged again to yield the final 25% NPs. Then, 25% NPs were resuspended in 10 mL of MES buffer, followed by the addition of EDC solution and NHS solution(Merck Millipore, Billerica, MA, USA). The mixture was stirred at 600 rpm for 3 h. The activated 25% NPs were collected by centrifugation at 18,000× *g* for 1 h and then resuspended in a solution containing recombinant proteins. The pH of the solution was adjusted to 8.0, and the mixture was stirred vertically overnight at 4 °C. The hydrodynamic diameter and zeta potential of 25% NPs, 25% NP-rHlaH35L, and 25% NP-rSpam were determined using dynamic light scattering (Delsa™ Nano, Beckman-Coulter, High Wycombe, UK). Fourier Transform Infrared Spectroscopy (Thermo Nicolet IS50, Thermo Fisher Scientific, Waltham, MA, USA) was utilized to verify successful crosslinking. The Bradford protein assay (Beyotime, Shanghai, China) was employed to measure the total recombinant protein concentration loaded onto 25% NPs, and the loading efficiency was calculated using the following formula:(1)$$Loading\;effiency\left(mass\;ratio\right)=\frac{m_{recombinant\;protein\;on\;the\;25\%NPs\left(mg\right)}}{m_{25\%NPs\left(mg\right)}}\times 100\%$$

### 2.3. Evaluation of Biocompatibility of 25% NP Antigen In Vitro and In Vivo

Cell counting kit-8 (CCK-8) was performed using the L929 fibroblast cell line to assess the cytotoxicity of nanovaccines. In brief, cells were plated at a concentration of 5 × 10^3^ cells/well in 96-well plates and allowed to adhere overnight. Gradient concentrations of nanovaccines, antigens, and nanoparticle solutions (ranging from 1000 μg/mL to 7.8125 μg/mL) were added to the wells, with each concentration tested in six replicate wells. After 24 h of incubation, CCK-8 solution was added to each well, followed by incubation at room temperature in the dark for 1 h. Absorbance was measured at 450 nm. Cell viability was determined using the following formula:(2)$$Cell\;Viability=\frac{A_{test}-A_{background}}{A_{control}-A_{background}}\times 100\%$$

Female BALB/C mice, aged 8 to 10 weeks, were randomly assigned to 6 groups and immunized subcutaneously with nanovaccines or antigens on day 0, followed by boost immunizations on day 14 and day 28 to evaluate the in vivo biocompatibility of the nanovaccines. The body weight and physical condition (including activity level, appetite, coat condition, and skin integrity) of the mice were recorded on days 0, 7, 14, 21, 28, and 35. On day 35, the mice were euthanized, and tissues from the heart, liver, spleen, lung, and kidney were collected. Hematoxylin and eosin (HE) staining, following the standard method described in our earlier publication [36], was performed to examine tissue toxicity caused by the nanovaccine.

### 2.4. Animal Immunization

BALB/C mice used in the experiments were purchased from the Animal Experiment Center of Air Force Medical University. All animal experiments complied with the National Research Council’s Guide for the Care and Use of Laboratory Animals, and the animal experiments were approved by the Animal Care and Ethics Committee of Fourth Military Medical University (Approval NO. KY20213144-1). Female BALB/C mice of 8–10 weeks of age were used for immunization, and all animals were healthy before immunization. Animals were divided into 7 groups and inoculated with PBS buffer, rHlaH35L, rSpam, rHlaH35L + rSpam, clinically used alum mixed with rHlaH35L (alum-rHlaH35L), rSpam (alum-rSpam), and alum-rHlaH35L + alum-rSpam were set as positive control, and 25% NP antigens were set as test groups. Mice were immunized subcutaneously on days 14 and 28 after the first immunization for a boost with the same formulation (antigen concentration of 25 μg/mice). Based on the average antigen loading efficiency and antigen dosage, the concentration of nanoparticles in 25% NP-rHlaH35L was 2272.7 μg/mice; the concentration of nanoparticles in 25% NP-rSpam was 1388.9 μg/mice; the concentration of nanoparticles in the antigen combine group was 3661.6 μg/mice.

### 2.5. ELISA Assay

Blood was collected from the tail tip vein of mice on the 17th and 35th days from the initial immunization and incubated at 37 °C for 1 h to allow the serum to precipitate. The serum was centrifuged and stored at −80 °C. The antigen was diluted to 10 μg/mL with coating buffer and added to a 96-well plate and stored at 4 °C overnight. Mouse serum was diluted to different gradients with antibody diluent solution. The gradient-diluted mouse serum was added to a 96-well plate and incubated at 37 °C for 1 h. Then, the plate was washed 5 times with PBST. HRP-labeled secondary antibody was added and incubated at 37 °C for 45 min, and then the plate was washed with PBST 5 times, and the plate was incubated at room temperature for 20 min with the addition of ABTS chromogenic solution. The absorbance at 405 nm was detected by a microplate reader (Biotek SYNERGY LX Instruments, Santa Clara, CA, USA). The AUC value of the antibody titer was calculated as we reported before.

### 2.6. ELISPOT Assay

Mice spleens were isolated and ground to prepare mice splenocyte homogenates after the mice were euthanized in an aseptic environment. The splenocyte homogenate was centrifuged, mixed with erythrocyte lysate, left at room temperature for 5 min, centrifuged again, and counted, and the cell density was adjusted to 1 × 10^7^ cells/mL. The 96-well plate was washed 5 times using sterile PBS and then incubated with 1640 medium containing 10% FCS at room temperature for 30 min. The antigen was diluted with medium to 4 μg/mL, and 100 μL of antigen solution was added to each well as an immunostimulant. Finally, 100 μL of cell suspension was added, and the plate was incubated for 24 h at 37 °C. The plate was washed 5 times with sterile PBS. The detection antibody was diluted to working concentration with 0.5% FCS in PBS. A total of 100 μL of detection antibody was added to each well and incubated for 2 h at room temperature. The plate was washed 5 times with sterile PBS, and 100 μL of ACP solution diluted with 0.5% FCS-PBS was added to each well and incubated for 1 h at room temperature. The plate was washed five times with sterile PBS, and 100 μL of TMB solution was added to each well until unique spots appeared. The plates were rinsed with plenty of water to terminate the color development and counted in an ELISPOT reader (Cellular Technology Limited, Cleveland, OH, USA).

### 2.7. Lethal Challenge

A total of 100 μL of *S. aureus* (strain ATCC 25923) at different concentrations (for the 1 × LD_100_ lethal challenge group, the concentration of *S. aureus* was 2.56 × 10^9^ CFU/mL; for the 2 × LD_100_ lethal challenge group, the concentration of *S. aureus* was 5.12 × 10^9^ CFU/mL; and for the 4 × LD_100_ lethal challenge group, the concentration of *S. aureus* was 1.024 × 10^10^ CFU/mL) were administered intravenously to the mice on the 35th day after prime immunization. Then, the physical condition and mortality of mice in each group were observed and recorded daily. At the end point of the experiment, all remaining animals were euthanized.

### 2.8. Serum Bactericidal Test

The Neisser–Wechsberg tube lysis reaction was used to determine the neutralizing activity of antibodies in the sera of immunized mice. *S. aureus* ATCC 25923 was cultured to OD 600 nm = 0.5. Healthy guinea pig sera were isolated and prepared. Bacterial liquid, guinea pig serum, and mouse serum immunized with nanovaccine were mixed with sterile PBS at a dilution ratio of 1:1000,000 and incubated at 37 °C for 1 h. The solution was spread on Luria–Bertani solid plates, and single clones on the plates were counted on the next day. Mouse sera from PBS and antigen groups were used as a negative control for simultaneous experiments. The lysis rate was calculated using the following formula:(3)$$Percentage\;of\;lytic\;bacteria(\%)=1-\frac{The\;CFU\;of\;treated\;group}{The\;CFU\;of\;control\;group}\times 100\%$$

### 2.9. Statistical Analysis

All results were expressed as mean ± standard deviation (SD) and statistically analyzed using GraphPad Prism 9.0 software. Data distribution was assessed using the Shapiro–Wilk test, which confirmed normality. Analysis of variance (one-way ANOVA) was used to determine the statistical significance of the differences. Log-rank test was used to determine the statistical significance of survival rates. Statistical difference was defined as * *p* < 0.05, ** *p* < 0.01, *** *p* < 0.001, and **** *p* < 0.0001, and ns (not significant) for *p* ≥ 0.05 with *p* < 0.05 considered statistically significant. 

## 3. Results

### 3.1. General Characterizations of 25% NPs Conjugated with Antigen

The molecular weight and purity of rHlaH35L and rSpam were confirmed by SDS-PAGE (Appendix A). The successful covalent binding of the two antigens to 25% NPs was verified by the presence of amide bonds in the Fourier Transform Infrared Spectroscopy (FTIR) spectra of conjugated NP samples (Figure 2a). The loading efficiencies of rHlaH35L and rSpam on 25% NPs were approximately 1~2% (mass ratio), respectively (Figure 2b).

The determination results of the hydrodynamic diameter and zeta potential for various batches of nanoparticles and nanovaccines can be found in the Appendix A. The representative average size of 25% NPs is 172.9 ± 6.963 nm (Figure 3a). As shown in Figure 3b,c, a small increase of ~20 nm in the average size of the NPs was observed after the antigen conjugation (25% NP-rHlaH35L:189.9 ± 9.531 nm; 25% NP-rSpam:201.5 ± 8.834 nm). Since the synthesized nanoparticles are being carboxyl-terminated, the zeta potential of 25% NPs is −17.917 ± 1.107 mV (Figure 3d). Additionally, after antigen conjugation, the zeta potential of both 25% NP-rHlaH35L (−8.897 ± 0.956 mV) and 25% NP-rSpam (−22.074 ± 1.898 mV) remains negative (Figure 3e,f).

### 3.2. Biocompatibility of Nanovaccines

The vaccine demonstrated good safety both in vitro and in vivo. In the CCK-8 experiments, the survival rate of the L929 cell line remained above 95% at nanoparticle concentrations ranging from 7.8 ng/mL to 1 mg/mL (Appendix A). The body weights of the immunized mice did not show any significant decrease during the monitoring period (Appendix A). The immunized mice were in good physiological condition, exhibiting normal appetites and high levels of vitality. Their skin and fur were in excellent condition. No significant damage was observed in the major organs of the immunized mice (Appendix A).

### 3.3. The Efficacy Evaluation of Nanovaccines

After S.C. vaccination of the antigen, alum/antigen and 25% NP/antigen conjugates, the antigen-specific IgG levels were monitored for all treatment groups. In our research, mice were first inoculated at day 0, then boosted at day 14 and day 28. Blood samples were collected on day 17 and day 35 in order to evaluate the specific IgG levels in all of the treatment groups (Figure 4a). Day 17 was selected as the sampling point because IgM, the dominant antibody subtype before the second vaccination, exhibits a transient peak during the early immune response and typically declines as the immune system shifts to IgG production. According to the immune response rules, protective IgG antibody levels should rise after the second vaccination. As expected, and as shown in Figure 4b, the PBS buffer group and antigen groups (rHlaH35L, rSpam, and rHlaH35L + rSpam) did not show significant antibody growth. On the contrary, the positive alum adjuvant control groups, the alum-rHlaH35L and alum-rSpam treatment groups, showed significant IgG increases compared with the corresponding antigen ones. The results above confirmed that the evaluation system runs normally and that the results are logical. Then, we analyzed the results of the experimental groups, including the combination of the alum adjuvants with antigen and the antigen combined with 25% NP adjuvants. The alum-rHlaH35L + alum-rSpam treatment group did not show any significant difference compared to the groups receiving alum adjuvant with antigen alone. However, in the 25% NP-adjuvanted immune groups, either combined with the single antigen (rHlaH35L or rSpam) or the antigen combination (rHlaH35L and rSpam), antigen-specific IgG titers significantly increased. Moreover, the 25% NP adjuvant conjugated with the antigen combination presented double IgG AUC compared with the conjugated with a single antigen. The results above indicated that even in the early stage in the immunization period, only 3 days after the second vaccination, 25% NPs serving as a vaccine adjuvant could elicit a high antibody titer level.

As time went on, on day 35, the 7th day after the third vaccination, the PBS group and negative free antigen groups (including the single antigen and antigen combinations) showed no significant IgG increase (Figure 4c), which is comparable with existing research [37]. In the positive-alum control groups, alum could significantly increase the rHlaH35L and the combined antigen (rHlaH35L and rSpam)-specific IgG titer; however, alum-rSpam did not present a higher antibody titer level compared with the antigen groups. The absolute IgG AUC value of alum-rSpam on day 35 is comparable with the one on day 17, indicating that the alum adjuvant is not able to elicit a strong rSpam-induced humoral immune response. In the 25% NP adjuvant vaccine groups, 25% NPs conjugated with single rHlaH35L could elicit a similar IgG titer level with alum-rHlaH35L. Notably, it is pleasing that the 25% NP adjuvant could strongly elicit an rSpam-specific antibody titer level, which the alum adjuvant could not. The IgG level of 25% NP-rSpam is significantly higher than that of the alum-rSpam and pure rSpam vaccination groups. Consequently, the titer level of 25% NPs conjugated with the rHlaH35L and rSpam antigen combination is much higher than that of the alum ones.

As the antibody titer is representative of the humoral immune response, the secretion levels of IL-4 in the spleen of immunized mice exhibit a similar trend (Figure 4d). Both in the single-antigen vaccine group and in the antigen-combination vaccine group, the number of IL-4 spots in the 25% NP adjuvant vaccine groups is much higher than that in the alum ones, indicating that for *S. aureus* recombinant protein rHlaH35L and rSpam, the adjuvanticity for eliciting a humoral immune response of 25% NPs is higher than that of alum.

On the other hand, the cellular immune response activated by specific vaccine groups was also evaluated by an ELISPOT assay. IFN-γ spots were employed for measuring cellular immune response activation. As shown in Figure 4e, vaccines with adjuvant groups including alum and 25% NPs could elicit significant cellular immune responses. Furthermore, all 25% NP vaccine groups exhibited a significantly higher number of IFN-γ spots compared to the alum-adjuvanted groups, irrespective of the antigen formulation (single antigen or antigen combination). The results indicate that in the cellular immune response activation aspect, 25% NPs showed stronger adjuvanticity.

Apart from the enhancement of humoral and cellular immune response, Th17-associated immune responses are another consideration in the establishment of an effective vaccine design against *S. aureus* infections. Given that IL-17A produced by Th17 cells is an essential cytokine in the primary defense against *S. aureus* infections, an ELISPOT assay of IL-17A was also carried out to determining whether the 25% NP vaccine could eradicate *S. aureus* in the host. As shown in Figure 4f, as the vaccine adjuvant, 25% NPs could elicit five to six times the IL-17A-specific spots compared with the alum adjuvant when combined with a single antigen. Moreover, 25% NPs conjugated with the antigen combination (rHlaH35L and rSpam) could induce an IL-17A level in the serum 19 times higher than that of the alum group, indicating that 25% NPs as the vaccine adjuvant could effectively induce Th17-associated immune responses.

Although the evidence of humoral, cellular, and Th17-associated immune response activation was found in the 25% NP vaccine groups, the neutralizing capability of the antibody is the crucial factor that determines the vaccine protection ability. Bacteriolysis evaluation was employed to investigate the bacteriolysis effects of the antibody produced by vaccination of the 25% NP single or combined antigen (with the corresponding alum vaccine groups as the control groups). The results (Figure 4g) reveal the significantly higher bacteria-neutralizing capability of antibodies produced by mice vaccinated with 25% NP vaccines over the alum ones. The results are consistent with the mice survival rates in the lethal challenge experiment.

### 3.4. Lethal Challenge

Before the lethal challenge, all of the mice were healthy. For the groups of mice treated with pure antigen/the antigen combination or PBS, all of the mice died following intravenous administration of a lethal dose of *S. aureus*. (Appendix A). A total of 20% of the mice survived in the alum-rSpam-treated group and 50% of the mice survived in the alum-rHlaH35L- and alum-rHlaH35L + alum-rSpam-treated groups. In comparison, the survival rates of the 25% NP vaccine groups with pure antigen and the antigen combination were much higher (about 100%) than those of the alum vaccine groups in general (Figure 5b–d). No significant difference was identified at this lethal challenge dose using the single-antigen or antigen-combination 25% NP vaccines. However, in the efficacy evaluation above, we did observe significant differences in terms of humoral, cellular, or Th17-associated immune response activations between the three 25% NP vaccine groups. The results suggest that the lethal dose is too low to differentiate the efficacy of these three nanovaccines. Thus, we increased the challenge dose to a 2 × LD_100_ and 4 × LD_100_ lethal dose.

As shown in Appendix A, the same situation happened in the mouse groups treated with pure antigen/the antigen combination or PBS; 100% mortality was observed in all mice after intravenous injection of a 2 × LD_100_ dose of *S. aureus*.". In the alum-adjuvant vaccine groups, all of the mice in the alum-with-a-single-antigen (rHlaH35L or rSpam)-treated groups died within one day after the challenge. Only 20% of the mice survived in the alum-rHlaH35L + alum-rSpam-treated group (Appendix A). However, the survival rates of the 25% NP vaccine group with pure antigen and the antigen combination were still much higher than those of the alum vaccine groups. A total of 100% of the mice survived in the 25% NP-rHlaH35L-treated group, while 40% and 80% of the mice survived in the 25% NP-rSpam and 25% NP-rHlaH35L + 25% NP-rSpam treated groups (Figure 6b). The addition of rSpam resulted in a decrease in survival percentage in the 25% NP vaccine groups. We further challenged all of the mouse groups with even higher doses.

As expected, all mice in the negative control and alum-adjuvanted groups died following intravenous administration of a 4 × LD_100_ dose of *S. aureus.* (Appendix A). For the 25%-NP-adjuvant-vaccine-treated groups, the survival rate of the mice treated with 25% NP-rSpam is significantly lower than those treated with rHlaH35L (Figure 6c).

The vaccine efficacy and lethal challenge results indicated that for the 25% NP adjuvant, rHlaH35L is a more probable antigen choice, with which 25% NPs could elicit higher neutralizing antibody titers and stronger body protection in mice.

## 4. Discussion

In recent years, the development of a vaccine for *S. aureus* has been recognized as a key strategy for controlling the prevalence of MRSA. However, to date, no human *S. aureus* vaccine has been successfully tested in clinical trials and commercialized. The critical role of adjuvants in *S. aureus* vaccines has been widely recognized. Due to their excellent biocompatibility and controlled drug release properties, PLGA nanoparticles have been regarded as a promising adjuvant in the development of *S. aureus* vaccines. Cui et al. developed an IsdB_137-361_-encapsulated PLGA-PEI nanoparticle vaccine, which could significantly promote the secretion of anti-infection cytokines and effectively reduce the bacterial loads in the main organs of immunized mice. Compared with the negative control group, the survival rate of the experimental group was significantly improved in the lethal challenge experiment [38]. Based on the foundation of our previous study, a PLGA nanoadjuvant with a specific size, shape, and rigidity (25% NPs) was developed in this study. Compared with the clinically used aluminum adjuvant, 25% NPs significantly elevated neutralizing antibody levels and improved the survival rate of mice in lethal challenge experiments. Vaccines containing 25% NPs as adjuvants have been demonstrated to induce the secretion of high levels of anti-infection cytokines, an effect that cannot be achieved by the alum adjuvant. Meanwhile, 25% NPs have tunable physicochemical properties and freeze-dried characteristics, which expands their potential applications.

The compatibility between antigen and adjuvant may be another vital factor affecting the efficacy of *S. aureus* vaccines, which is less discussed. Mahdavi et al. prepared vaccines using PLGA nanoparticles as adjuvants and PBP-2A protein with MRSA autolysin as antigens. The antigen exerted an effect on the survival rate of mice in the lethal challenge: all of the mice immunized with r-PBP2a-PLGA survived, whereas the survival rate of the mice immunized with r-autolysin-PLGA was 90% [39]. This result suggests that PLGA nanoparticles may have a potential antigen preference.

Based on the principle of the biomimetic design of adjuvants and the pathogenic mechanism of *S. aureus*, the secretion characteristics of *S. aureus* virulence factors may be a pivotal factor influencing adjuvant preference. The majority of exotoxins are soluble proteins secreted extracellularly by *S. aureus*, which can diffuse into the host environment and exert their toxic effects over long distances. Exotoxins are mainly involved in the process of bacterial virulence regulations, whereas cell-wall-anchored proteins are linked to the bacterial cell wall through covalent or non-covalent binding and directly interact with the host cells or immune components. They are involved in the processes of bacterial adhesion and colonization, immune escape, and biofilm formation. The mechanisms of the exotoxin Hla and cell-wall-anchored protein Spa in *S. aureus* infection have been intensively studied and have been widely used in the development of *S. aureus* vaccines. However, there are limitations to these studies. Burns et al. immunized mice with HlaH35L in combination with alum adjuvants and CpG adjuvants. The vaccine significantly reduced the bacterial loads in the kidneys of the mice. However, only one of the two experiments yielded statistically significant results, and the reproducibility of the results needs to be further examined [30]. Schneewind et al. immunized mice with a mixture of HlaH35L and Freund’s adjuvant. Lethal challenge experiments were conducted using *S. aureus* strains NEWMAN, LAC, and MW2 [40]. Survival rates were significantly increased in mice. However, this study did not examine the cellular response elicited by vaccines. Schneewind examined the protective ability of the SpA _KKAA_ mutant mixed with Freund’s adjuvant [41]. The SpA _KKAA_-immunized mice produced a higher level of specific antibodies and displayed higher survival rates in lethal challenge experiments. Additionally, the bacterial loads and abscess lesions in the kidney tissues of SpA _KKAA_-inoculated mice were substantially reduced. But again, this study did not investigate the effect of the SpA _KKAA_ mutant on cellular immunity.

The present study examined the preference of 25% NPs on rHlaH35L and rSpam from multiple dimensions. As demonstrated in the Results section, 25% NP-rHlaH35L induced higher levels of antibody titers and antimicrobial cytokine secretion levels compared to 25% NP-rSpam and significantly increased the survival rate of mice in high-dose lethal challenge experiments. We speculated that the number of neutralizing antibodies might be one of the key factors leading to the development of antigenic preferences in 25% NPs. Although the lytic capacity of serum antibodies in mice in the 25% NP-rHlaH35L and 25% NP-rSpam groups showed similar trends in the in vitro experiments, the antibody titers of mice in the 25% NP-rHlaH35L group were higher than those of the 25% NP-rSpam group. This indicates that more antibodies with neutralizing activity were produced in the mice in the 25% NP-rHlaH35L group. In addition, mice in the 25% NP-rHlaH35L group exhibited elevated levels of IL-4, IL-17A, and IFN-γ cytokine secretion compared with those in the 25% NP-rSpam group. Elevated levels of IL-4 suggest activation of Th2-type immune responses, while a significant increase in IL-17A reflects activation of Th17 cells, and elevated levels of IFN-γ indicate enhanced Th1-type immune response. This synergistic effect of multidimensional immune response may enhance the body’s ability to fight infections by mechanisms such as enhancing antigen presentation, promoting effector T cell differentiation, and strengthening immune memory. The specific mechanisms need to be further investigated.

The present study also has the following limitations. First, despite the use of an intravenous MRSA model to simulate systemic infections, the pathologic process still differs significantly from that of clinical patients. *S. aureus* often achieves persistence in the human body through biofilm-mediated chronic infections (e.g., artificial joint infections) or repeated colonization in immunosuppressed hosts, and it is challenging to reproduce this complex microenvironment of multistage, multiorgan interactions in the acute attack model. Second, there is room for optimization of antigen screening strategies. The current study only focused on the secretion characteristics of bacterial virulence factors, and only two representative antigens were selected. In the future, new antigens need to be introduced and combined with deep-learning-driven epitope prediction to select antigen peptides matching the physicochemical properties of the PLGA adjuvant in order to break through the current bottleneck of antigen selection. Third, the surface functionalization of PLGA adjuvants has not been fully explored. Existing nanoparticles rely on passive targeting strategies, but the introduction of active targeting elements (e.g., CD11c single-chain antibody fragment modification to enhance dendritic cell uptake) or immune signaling synergistic modules (e.g., the covalent coupling of CpG with a TLR9 agonist) can build an integrated delivery/activation smart adjuvant. Finally, the study did not address the potential variability in immune responses between different populations, such as immunocompromised individuals or those with pre-existing conditions. Future studies should include diverse experimental groups to evaluate the broader applicability of the findings.

## 5. Conclusions

In conclusion, this work initially explores the selective preference of PLGA nanoparticles with a 25% PEG doping ratio to *S. aureus* antigens with different secretion characteristics. Compared with rSpam, a cell-wall-anchor protein, a nanovaccine composed of 25% NPs and rHlaH35L (an HLA-modified recombinant antigen of a *Staphylococcus aureus* exotoxin) induced higher levels of neutralizing antibody titer and anti-infective cytokine secretion. In addition, the 25% NP-rHlaH35L vaccine provided more efficient immune protection for mice in the lethal challenge experiment with large doses of bacteria, suggesting that in the development of the recombinant *S. aureus* vaccine, in addition to reasonable screening of vaccine components, the selective preference of adjuvants for antigens, which is antigen-dependent adjuvanticity, may be a pivotal factor affecting vaccine efficacy.

## Figures and Tables

**Figure 1 vaccines-13-00317-f001:**
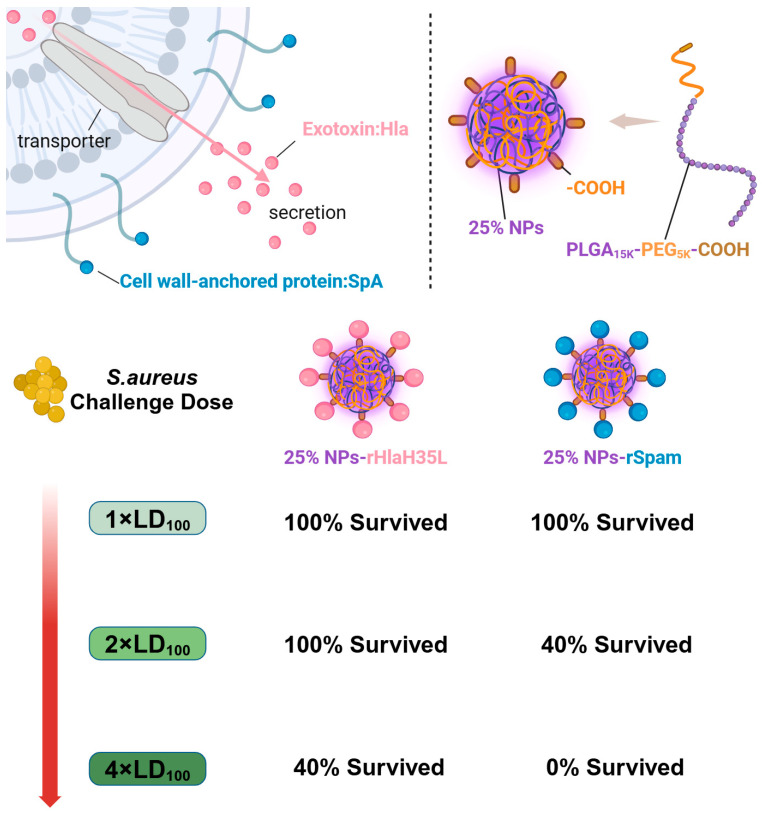
A graphic illustration of the immune efficacy of antigen-dependent adjuvants. Nanovaccines prepared with 25% NPs with either the exotoxin Hla or the CWA protein SpA exhibited different protection at different challenge doses. When the challenge dose was 1 × LD_100_, both nanovaccines provided 100% protection. However, when the challenge dose increased to 2 × LD_100_, the survival rate of 25% NP-rHlaH35L remained 100%, while the survival rate of the 25% NP-rSpam group was 40%. As the challenge dose continued to increase, the survival rates of both groups decreased, and the protective efficacy of 25% NP-rHlaH35L was superior to that of the 25% NP-rSpam group. Created in BioRender (https://BioRender.com).

**Figure 2 vaccines-13-00317-f002:**
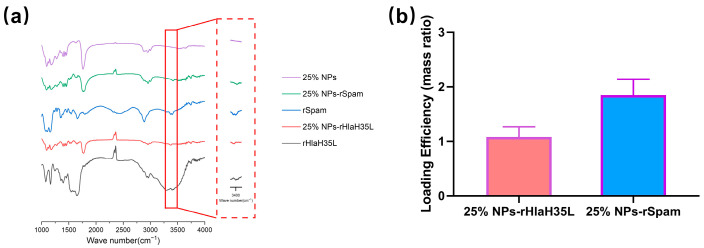
Nanovaccine via condensation reaction as enhanced *S. aureus* vaccine adjuvant. (**a**) FTIR spectra taken from 25% NPs, 25% NP-rHlaH35L, rHlaH35L, 25% NP-rSpam, and rSpam samples. (**b**) Antigen loading efficiency of 25% NP antigen measured by Bradford.

**Figure 3 vaccines-13-00317-f003:**
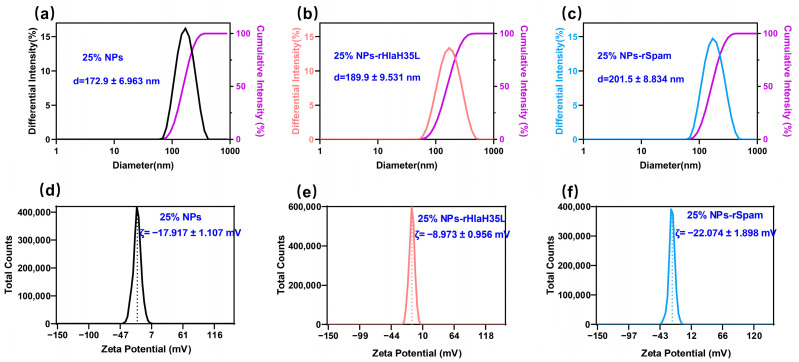
Hydrodynamic diameter of (**a**) 25% NPs, (**b**) 25% NP-rHlaH35L, and (**c**) 25% NP-rSpam. Zeta potential of (**d**) 25% NPs, (**e**) 25% NP-rHlaH35L, and (**f**) 25% NP-rSpam.

**Figure 4 vaccines-13-00317-f004:**
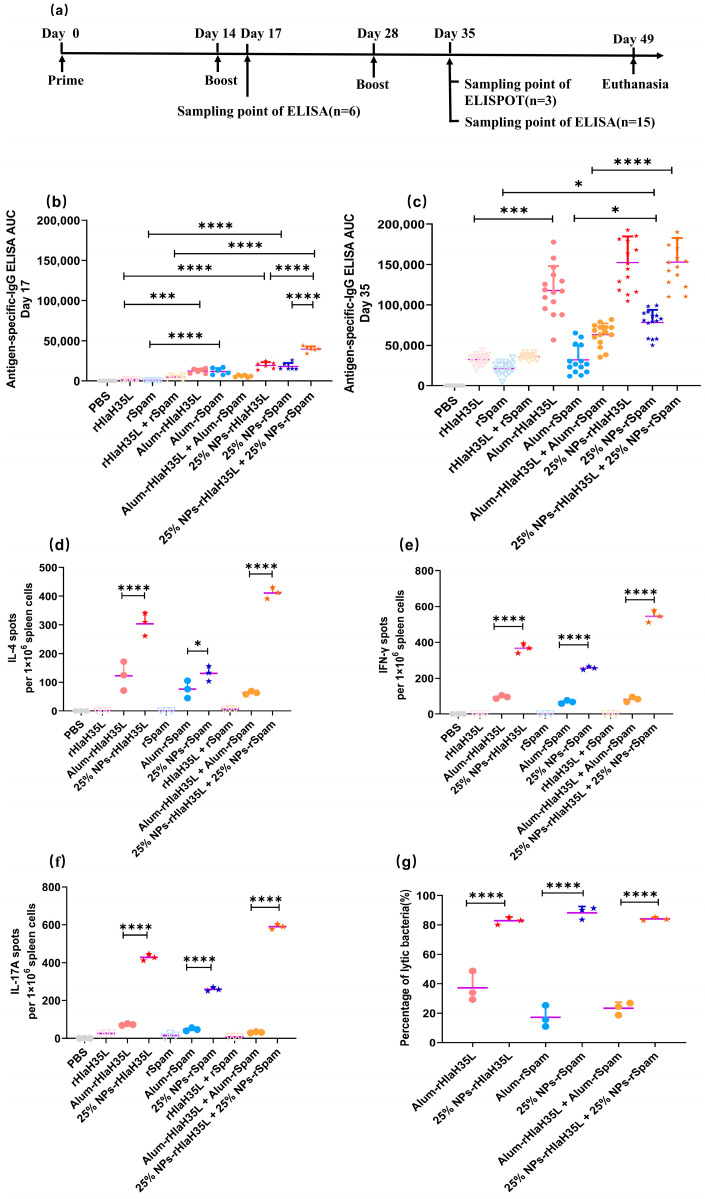
Immune activation response elicited by vaccines. (**a**) Diagram of immunization procedure and sampling time points for immunoassay. (**b**) Antigen-specific serum IgG titer at day 17 (*n* = 6). (**c**) Antigen-specific serum IgG titer at day 35 (*n* = 15). (**d**) IL-4 secretion level revealed by ELISPOT at day 35 (*n* = 3). (**e**) IFN-γ secretion level revealed by ELISPOT at day 35 (*n* = 3). (**f**) IL-17A secretion level revealed by ELISPOT at day 35 (*n* = 3). (**g**) Percentage of lytic bacteria of serum antibody at day 35. All data are expressed as mean ± S.D. Statistical significance is indicated by *p* < 0.05 (* *p* < 0.05; *** *p* < 0.001; **** *p* < 0.0001).

**Figure 5 vaccines-13-00317-f005:**
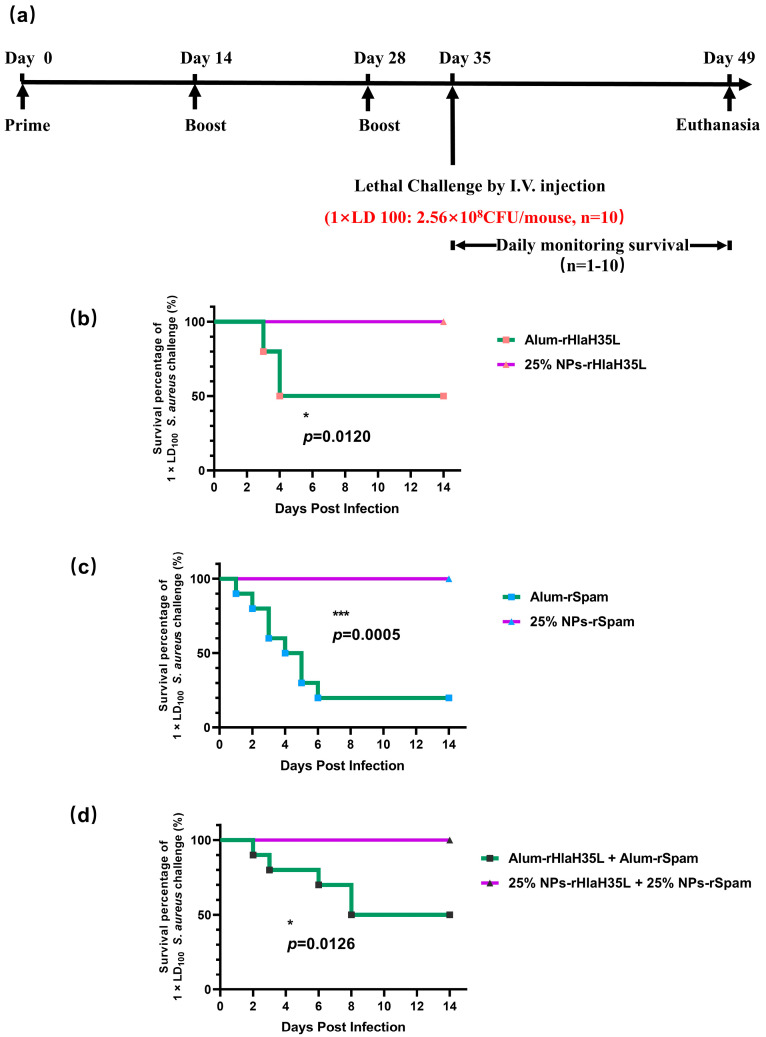
Survival analysis after lethal challenge with *S. aureus* strain ATCC 25923 at concentration of 2.56 × 10^8^ CFU per mouse (**a**) Diagram of immunization, lethal challenge, and evaluation of protective efficacy. (**b**) Survival analysis between mice immunized with alum-rHlaH35L and mice immunized with 25% NP-rHlaH35L (*n* = 10). (**c**) Survival analysis between mice immunized with alum-rSpam and mice immunized with 25% NP-rSpam (*n* = 10). (**d**) Survival analysis between mice immunized with alum-rHlaH35L + alum-rSpam and mice immunized with 25% NP-rHlaH35L + 25% NP-rSpam (*n* = 10). Survival rates were analyzed using Log-rank (Mantel–Cox) analysis. Statistical significance is indicated by *p* < 0.05 (* *p* < 0.05; *** *p* < 0.001).

**Figure 6 vaccines-13-00317-f006:**
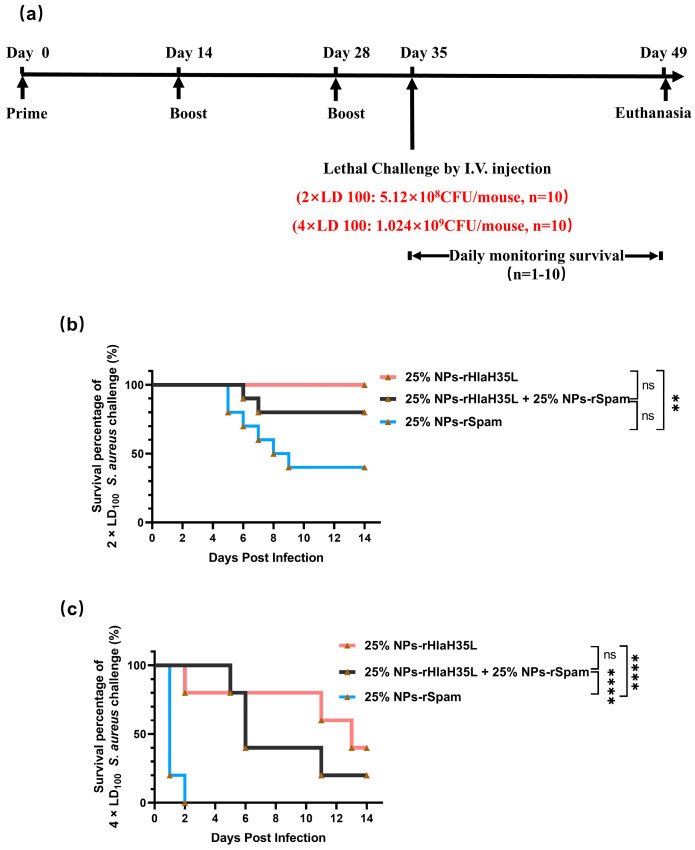
Survival analysis after lethal challenge with *S. aureus* strain ATCC 25923 at different bacteria concentrations. (**a**) Diagram of immunization, lethal challenge, and evaluation of protective efficacy. (**b**) Survival analysis among mice immunized with 25% NP-rHlaH35L, 25% NP-rSpam, and 25% NP-rHlaH35L + 25% NP-rSpam when mice were challenged with *S. aureus* strain ATCC 25923 at concentration of 5.12 × 10^8^ CFU per mouse (*n* = 10). (**c**) Survival analysis among mice immunized with 25% NP-rHlaH35L, 25% NP-rSpam, and 25% NP-rHlaH35L + 25% NP-rSpam when mice were challenged with *S. aureus* strain ATCC 25923 at concentration of 1.024 × 10^9^ CFU per mouse (*n* = 10). Survival rates were analyzed using Log-rank (Mantel–Cox) analysis. Statistical significance is indicated by *p* < 0.05 (** *p* < 0.01, **** *p* < 0.0001).

## Data Availability

Data will be made available upon request.

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
