# Peer review of "Antigen-Dependent Adjuvanticity of Poly(lactic-co-glycolic acid)-polyethylene Glycol 25% Nanoparticles for Enhanced Vaccine Efficacy"

_vaccines, 2025, doi:10.3390/vaccines13030317_

Round 1

Reviewer 1 Report

Comments and Suggestions for Authors

This is a work with a ton of experiments. These are my comments: 

Line 41, "when it conjugated"?

Line 42-43, "with maximized potency and minimized side effects.", within your study.

Line 86-88, "Spherical particles with a size of 193 nm produced a TH1-biased immune response, while larger rod-shaped particles (1530 nm) produced a stronger TH2-biased immune response", indicate the type of material, gold?

Line 93-96, "In short, a nanovaccine prepared with 25% 93 PEG-doped PLGA nanoparticles combined with the S. aureus antigen rEsxB provided 100% protection in a mouse S. aureus infection model during subcutaneous immunization[22]." You must talk about works from other authors using PLGA nanoparticles. 

Figure 1 seems to not be included in the text. 

Line 150, "coil"?

Line 151, "rHlaH35L and rSpam" you need to provide details on these antigens, are they whole proteins? are they regularly used and reported elsewhere?

Line 217-218, "clinically used Alum mixed with rHlaH35L(Alum-rHlaH35L), 217 rSpam (Alum-rSpam) and Alum-rHlaH35L+ Alum-rSpam", what was the dosage of alum?

Line 254, "100 μL of S. aureus", what was the cell concentration?

Line 287-291, "The average sizes of the 25%NPs are 172.9 nm (Fig. 3a), and a small increase of ~20 287 nm in the average size of the NPs was found after the antigen conjugation (Fig. 3b;3c). 288 Due to the synthesized nanoparticles is carboxyl terminated, the zeta potential of 25%NPs 289 is ~ -17 mV (Fig. 3d). Also, after antigen conjugation, the zeta potential of both 25%NPs-290 rHlaH35L and 25%NPs-rSpam is appeared to be negative (-8 mV~-22 mV)", you need to provide average plus minus standard deviation of all measurements (size and zeta potential), and they have to be obtained from different nanoparticle preparations, I am not talking about the standard deviation from the measurement per se. 172.9+-6.693 does not come from different preparations. 

Line 291, "appeared"? it is or it is not

Line 291, "(-8 mV~-22 mV)", calculate the theoretical pI of the recombinant proteins to establish that the zeta potential must remain negative.

 Line 341-342, "No matter in single antigen"?

Line 442, "series"?

Discussion and Conclusions, the discussion is poor in terms of comparing your results with the literature using the same or similar antigens and/or the same or similar nanoparticles. 

Comments on the Quality of English Language

There are typos throughout the document. The grammar can also improve.  

Author Response

Dear reviewer,

We are very grateful for your constructive comments and suggestions for our manuscript entitled“Antigen dependent adjuvanticity of PLGA-PEG 25%NPs for enhanced vaccine efficacy.”(ID:3454176). Your comments are very valuable and helpful for improving our manuscript. In the following, the response to all the comments have been provided one by one.

We have tried our best to make all the revisions clear and we hope that the revised manuscript can satisfy the requirements for publication.

Sincerely,

Li Fan; ChaoJun Song.

Comments 1: Line 41, "when it conjugated"?

Response 1: In order to avoid the confusion, the sentence "when it conjugated" has been corrected as "when different kinds of antigens are conjugated". Please see line 42-43 in the revised manuscript.

Comments 2: Line 42-43, "with maximized potency and minimized side effects.", within your study.

Response 2: We sincerely thank the reviewer for their insightful comment. We believe that the phrase "with maximized potency and minimized side effects" may appear overly absolute. To address this concern, we have replaced "maximized" with "enhanced" and "minimized" with "reduced" to better align with the experimental results and avoid overstatement. Please see line 44 in the revised manuscript.

Comments 3: Line 86-88, "Spherical particles with a size of 193 nm produced a TH1-biased immune response, while larger rod-shaped particles (1530 nm) produced a stronger TH2-biased immune response", indicate the type of material, gold?

Response 3: Thank you for raising this question and we apologize for the confusion. Polystyrene nanoparticles are used in this reference to illustrate how physiochemical parameter such as size and shape influence the type and degree of vaccine immune activation. We have added the type of material. Please see line 86-87 in the revised manuscript.

Comments 4: Line 93-96, "In short, a nanovaccine prepared with 25% 93 PEG-doped PLGA nanoparticles combined with the S. aureus antigen rEsxB provided 100% protection in a mouse S. aureus infection model during subcutaneous immunization [22]." You must talk about works from other authors using PLGA nanoparticles.

Response 4: We sincerely thank the reviewer for valuable suggestion. As recommended, we have amended the corresponding content to include relevant studies from other authors using PLGA nanoparticles Please see line 96-107 in the revised manuscript.

Comments 5: Figure 1 seems to not be included in the text.

Response 5: Please see line 133-142 in the revised manuscript and below.

Figure 1. Graphic Illustration of the immune efficacy of antigen-dependent adjuvants. Nanovaccines prepared by 25% NPs with either exotoxin Hla or CWA protein SpA exhibited different protection in different challenge doses. When the challenge dose was 1×LD100, both nanovaccines provided 100% protection. However, when the challenge dose increased to 2×LD100, the survival rate of 25%NPs-rHlaH35L remained 100%, while the survival rate of the 25%NPs-rSpam group was 40%. As the challenge dose continued to increase, the survival rates of both groups decreased, and the protective efficacy of 25%NPs-rHlaH35L was superior to that of the 25%NPs-rSpam group. Created in BioRender. https://BioRender.com.

Comments 6: Line 150, "coil"?

Response 6: Thank you for raising this question, we feel sorry for our careless mistake. Spelling error “coil” has been corrected to “coli” in the revised manuscript. Please see line 156 in the revised manuscript.

Comments 7: Line 151, "rHlaH35L and rSpam" you need to provide details on these antigens, are they whole proteins? are they regularly used and reported elsewhere?

Response 7:

Thank you for pointing this problem in the manuscript and we are sorry for our negligence of providing details of recombinant protein rHlaH35L and rSpam. The detail of two recombinant protein have been added. Please see line 145-155 in the revised manuscript.

α-hemolysin(Hla) is an exotoxin of Staphylococcus aureus which plays an vital role in the Staphylococcus aureus infection. Hla binds to the ADAM-10 receptor on the cell membranes of target cells (rabbit red blood cells, human platelet cells, endothelial cells, epithelial cells, and leukocytes) and subsequently oligomerizes into heptamers. These heptamers extend into the phospholipid bilayer of the cell membrane, forming mushroom-shaped β-barrel structures that create hydrophilic transmembrane pores, ultimately leading to the lysis of target cells. In other words, Hla exhibits hemolytic activity. To remove the hemolytic toxicity of Hla, the histidine at position 35 in the Hla sequence was mutated to leucine to obtain the HlaH35L plasmid. HlaH35L has been used as model antigen in the development of Staphylococcus aureus vaccine and reports can be found as below:

1.Teymournejad, O.; Li, Z.; Beesetty, P.; Yang, C.; Montgomery, C.P. Toxin expression during Staphylococcus aureus infection imprints host immunity to inhibit vaccine efficacy. NPJ Vaccines 2023, 8, 3, doi:10.1038/s41541-022-00598-3.

2.Wardenburg, J.B.; Schneewind, O. Vaccine protection against Staphylococcus aureus pneumonia. Journal of Experimental Medicine 2008, 205, 287-294, doi:10.1084/jem.20072208.

  1. Brady, R.A.; Mocca, C.P.; Prabhakara, R.; Plaut, R.D.; Shirtliff, M.E.; Merkel, T.J.; Burns, D.L. Evaluation of genetically inactivated alpha toxin for protection in multiple mouse models of Staphylococcus aureus infection. PloS one 2013, 8, e63040, doi: 10.1371/journal.pone.0063040.
  2. Zuo, Q.-F.; Yang, L.-Y.; Feng, Q.; Lu, D.-S.; Dong, Y.-D.; Cai, C.-Z.; Wu, Y.; Guo, Y.; Gu, J.; Zeng, H.; et al. Evaluation of the Protective Immunity of a Novel Subunit Fusion Vaccine in a Murine Model of Systemic MRSA Infection. PloS one 2013, 8, e81212, doi: 10.1371/journal.pone.0081212.
  3. Irene, C.; Fantappiè, L.; Caproni, E.; Zerbini, F.; Anesi, A.; Tomasi, M.; Zanella, I.; Stupia, S.; Prete, S.; Valensin, S.; et al. Bacterial outer membrane vesicles engineered with lipidated antigens as a platform for Staphylococcus aureus vaccine. Proceedings of the National Academy of Sciences of the United States of America 2019, 116, 21780-21788, doi:10.1073/pnas.1905112116.

Protein A (SpA) is a cell wall-anchored protein expressed by Staphylococcus aureus and is a key protein in the bacterial immune evasion process. SpA is composed of five highly conserved immunoglobulin-binding domains (E, D, A, B, and C) that are arranged in tandem. These domains are followed by a highly variable region known as the X region. The C-terminal LPXTG motif of the X region mediates the covalent anchoring of SpA to the bacterial cell wall. The immunoglobulin-binding domains (E, D, A, B, and C) of SpA are commonly utilized as antigens in Staphylococcus aureus vaccines. Two consecutive glutamines (QQ) at positions 7-8 and two consecutive aspartic acid (AA) at positions 34-35 in the five structural domains in wild Spa are essential for the immune escape function of Spa. In order to obtain a non-toxic spa, the glutamine at positions 7-8 in each of the structural domains was mutated to lysine, and the aspartic acid at positions 35-36 was mutated to alanine. The mutated SpA was named Spam. Reports can be found as below:

  1. Kim Hwan, K.; Emolo, C.; DeDent Andrea, C.; Falugi, F.; Missiakas Dominique, M.; Schneewind, O. Protein A-Specific Monoclonal Antibodies and Prevention of Staphylococcus aureus Disease in Mice. Infection and Immunity 2012, 80, 3460-3470, doi:10.1128/iai.00230-12.
  2. Thammavongsa, V.; Rauch, S.; Kim, H.K.; Missiakas, D.M.; Schneewind, O. Protein A-neutralizing monoclonal antibody protects neonatal mice against Staphylococcus aureus. Vaccine 2015, 33, 523-526, doi: https://doi.org/10.1016/j.vaccine.2014.11.051.
  3. Zeng, H.; Yang, F.; Feng, Q.; Zhang, J.; Gu, J.; Jing, H.; Cai, C.; Xu, L.; Yang, X.; Xia, X.; et al. Rapid and Broad Immune Efficacy of a Recombinant Five-Antigen Vaccine against Staphylococcus aureus Infection in Animal Models. Vaccines 2020, 8, doi:10.3390/vaccines8010134.

Comments 8: Line 217-218, "clinically used Alum mixed with rHlaH35L(Alum-rHlaH35L), 217 rSpam (Alum-rSpam) and Alum-rHlaH35L+ Alum-rSpam", what was the dosage of alum?

Response 8: Thank you for raising this valuable question. Alum adjuvant used in this research were purchased from Croda. (Alhydrogel® adjuvant 2%, Catalog code: vac-alu-50). According to the User Guide: “Aluminum content is 9.0-11.0 mg/mL while the final volume ratio of Alhydrogel® adjuvant 2% to antigen should be 1:1 to 1:9.” We chose 1:1(v/v) so the final concentration of Alum was 5 mg/mL. The volumes for S.C. injection was 200 μL per mouse. So the dosage of Alum in Alum-rHlaH35L group and Alum-rSpam was 1 000 μg, respectively while the dosage in Alum-rHlaH35L+ Alum-rSpam group was 2 000 μg in total.

Comments 9: Line 254, "100 μL of S. aureus", what was the cell concentration?

Response 9: Thank you for your considerable attention. The cell concentration differs in three lethal challenge groups. To be specific, for 1×LD 100 lethal challenge group,the concentration of S. aureus was 2.56×109 CFU/mL. For 2×LD 100 lethal challenge group,the concentration of S. aureus was 5.12×109 CFU/mL while for 4×LD 100 lethal challenge group,the concentration of S. aureus was 1.024×1010 CFU/mL. The cell concentration in different lethal challenge groups has been added. Please see line 259-263 in the manuscript.

Comments 10: Line 287-291, "The average sizes of the 25%NPs are 172.9 nm (Fig. 3a), and a small increase of ~20 287 nm in the average size of the NPs was found after the antigen conjugation (Fig. 3b;3c). 288 Due to the synthesized nanoparticles is carboxyl terminated, the zeta potential of 25%NPs 289 is ~ -17 mV (Fig. 3d). Also, after antigen conjugation, the zeta potential of both 25%NPs-290 rHlaH35L and 25%NPs-rSpam is appeared to be negative (-8 mV~-22 mV)", you need to provide average plus minus standard deviation of all measurements (size and zeta potential), and they have to be obtained from different nanoparticle preparations, I am not talking about the standard deviation from the measurement per se. 172.9+-6.693 does not come from different preparations.

Response 10: Thank you for your constructive feedback on our manuscript and we appreciate it very much. In response to the reviewer's comment, we have conducted comprehensive characterization of the nano-adjuvants and nano-vaccines by measuring their size and zeta potential across multiple preparation batches. These detailed experimental data have been documented in the supporting information (Table 1 and Table 2) and below. Furthermore, we have incorporated the most representative results within the revised manuscript. Please see line 298-304.

Table 1. Hydrodynamic diameter of nanoparticles and nano-vaccines.

25%NPs(nm)

25%NPs-rHlaH35L(nm)

25%NPs-rSpam(nm)

Batch 1

174.2±4.44

200.1±7.92

202.1±6.77

Batch 2

169.1±6.21

187.2±6.21

191.7±4.12

Batch 3

177.5±8.12

193.1±7.89

196.2±7.23

Batch 4

172.9±6.95

189.9±9.53

201.5±8.834

Batch 5

170.7±5.12

196.2±5.72

205.6±8.12

Batch 6

173.6±4.92

192.0±4.22

203.9±6.54

Table 2. Zeta potential of nanoparticles and nano-vaccines.

25%NPs(mV)

25%NPs-rHlaH35L(mV)

25%NPs-rSpam(mV)

Batch 1

-17.917±1.107

-8.974±0.956

-22.074±1.898

Batch 2

-16.923±1.037

-8.265±0.385

-23.183±2.462

Batch 3

-18.211±2.000

-7.992±0.826

-20.362±3.018

Batch 4

-17.892±0.925

-7.362±1.036

-21.684±2.872

Batch 5

-16.275±1.673

-9.379±1.623

-22.569±1.992

Batch 6

-18.209±2.836

-9.002±1.362

-21.473±1.274

Comments 11: Line 291, "appeared"? it is or it is not

Response 11: Thank you for raising this question and we are sorry for this ambiguous expression. We have rephrased this sentence to make it clearer and more precise. Please see line 302-304 in the revised manuscript.

Comments 12: Line 291, "(-8 mV~-22 mV)", calculate the theoretical pI of the recombinant proteins to establish that the zeta potential must remain negative.

Response 12: Thank you for raising this question. We have calculated the theoretical pI of the two recombinant proteins. The theoretical pI of rHlaH35L is 7.93 while that of rSpam is 9.31. Due to the differences of the conjugated ratio and the specific pI of the proteins, the zeta potential of the vaccines shows specific negative charge as mentioned in the manuscript.

Comments 13: Line 341-342, "No matter in single antigen"?

Response 13: Thank you for raising this question and we are sorry for this confusion made by our faulty wording. We have amended the sentence “No matter in single antigen or the antigen combination conjugated with adjuvant vaccine group” into “Both in the single antigen vaccine group and in the antigen combination vaccine group”. Please see line 356-357 in the revised manuscript.

Comments 14: Line 442, "series"?

Response 14: Thank you for raising this question and we are sorry for this erroneous expression. We have revised the Discussion sections and deleted the imprecise expressions.

Comments 15: Discussion and Conclusions, the discussion is poor in terms of comparing your results with the literature using the same or similar antigens and/or the same or similar nanoparticles.

Response 15: We sincerely thank the reviewer for their insightful comment. We agree that the discussion section could be improved by providing a more detailed comparison of our results with those reported in the literature using similar antigens and/or nanoparticles. To address this, we have revised the Discussion sections to include a comprehensive comparison with relevant studies.

Comments 16: There are typos throughout the document. The grammar can also improve.

Response 16: Thank you for your careful reading and valuable feedback. We acknowledge that there were typos and grammatical errors throughout the manuscript. We have now thoroughly proofread the entire document and corrected all identified typos and grammatical issues.

Reviewer 2 Report

Comments and Suggestions for Authors

In the abstract, the authors state that they are interested in how the antigen affects adjuvanticity of NPs, stated as “antigen-dependent adjuvanticity”. This is basically asking how affective are different antigens, however, the data is really a comparison of different adjuvants (Alum, vs NPs) and then 2 antigens.

Abstract: The sentence “Challenging lethal experiments with different doses of bacteria were conducted to investigate the protection of the nano-vaccine” is unclear in meaning.

Introduction: The lit review cites studies that have shown a cell-mediated response is essential for S. aureus vaccines. The study compares alum with NPS and the data indicates that the greater cell-mediated response using NPs made for a better adjuvant, however, cell dependence was not shown beyond cytokine analysis.

Methods: What metrics are you using for physical condition (put in methods section). Also, I would say that mice were euthanized, rather than “executed”.

What is “regular vaccination period was adopted” based on?

What is an “ELISPOT histogram of IL-4 secretion”?

Figure 4 is fuzzy, with small print that is difficult to read.

Figure 5. Why are alum and NPs not included in the same studies?

Figure 6 & 7. Why are different antigens compared in different studies rather than the same study?

Fig. S2a what cell type was viability tested in? What assay was used?

Comments on the Quality of English Language

Grammar needs to be improved. Also, parts of the manuscript were unclear based on word choices.

Author Response

Dear reviewer,

We are very grateful for your constructive comments and suggestions for our manuscript entitled“Antigen dependent adjuvanticity of PLGA-PEG 25%NPs for enhanced vaccine efficacy.”(ID:3454176). Your comments are very valuable and helpful for improving our manuscript. In the following, the response to all the comments have been provided one by one.

We have tried our best to make all the revision clear and we hope that the revised manuscript can satisfy the requirements for publication.

Sincerely,

Li Fan; ChaoJun Song

Comments 1: In the abstract, the authors state that they are interested in how the antigen affects adjuvanticity of NPs, stated as “antigen-dependent adjuvanticity”. This is basically asking how affective are different antigens, however, the data is really a comparison of different adjuvants (Alum, vs NPs) and then 2 antigens.

Response 1:

We sincerely thank the reviewer for their careful reading of our manuscript and for raising this important point. We agree that the phrase “antigen-dependent adjuvanticity” may have caused confusion, as it could be interpreted as focusing solely on the effect of antigens on the adjuvanticity of nanoparticles (NPs). However, our study was designed to compare the protective efficacy of two different antigens (rHlaH35L and rSpam) formulated with PLGA-PEG 25%NPs as an adjuvant, using alum as a positive control. The primary goal was to evaluate how the combination of different antigens with the same adjuvant (PLGA-PEG 25% NPs) influences vaccine efficacy, rather than directly comparing the adjuvanticity of PLGA NPs and Alum. To address the reviewer’s concern and improve clarity, we have revised the abstract as follows:

Original Text:

“Nonetheless, how do various antigen selection affect the adjuvanticity of the nanoparticles is less discussed, which is important for practical application.”

Revised Text:

“Nonetheless, how the selection of different antigens influences the overall vaccine efficacy when combined with the same nanoparticle adjuvant is less discussed, which is important for practical applications.” Please see line 21-23 in the revised manuscript.

Original Text:

“Moreover, due to the antigen-dependent adjuvanticity, the animals inoculated with 25%NPs-rHlaH35L survived even after 2 folds of lethal dose of S. aureus administration.”

Revised Text:

“Moreover, due to the superior immune response elicited by 25%NPs-rHlaH35L, the animals inoculated with this formulation survived even after two times the lethal dose of S. aureus administration.” Please see line 38-41 in the revised manuscript.

Comments 2: Abstract: The sentence “Challenging lethal experiments with different doses of bacteria were conducted to investigate the protection of the nano-vaccine” is unclear in meaning.

Response 2:

We appreciate the reviewer’s comment regarding the clarity of the sentence. To better convey our experimental design, we have revised the sentence as follows: “To evaluate the protective efficacy of the nano-vaccine, immunized mice were challenged with S. aureus ATCC 25923 at three different lethal doses: 1×LD 100, 2×LD 100, and 4×LD 100.” Please see line 27-29 in the revised manuscript.

In this study, we used three escalating lethal doses of S. aureus ATCC25923 (1×LD100, 2×LD100, and 4×LD100) to rigorously assess the protective efficacy of the nano-vaccine. The LD100 (100% lethal dose) represents the minimum bacterial dose required to cause 100% mortality in unvaccinated control mice. By challenging immunized mice with 1×LD100, 2×LD100, and 4×LD100, we aimed to evaluate the robustness of the vaccine-induced protection under varying levels of bacterial exposure. This approach allows us to determine not only whether the vaccine provides protection but also the extent of protection under increasingly severe infection conditions.

Comments 3: Introduction: The lit review cites studies that have shown a cell-mediated response is essential for S. aureus vaccines. The study compares alum with NPS and the data indicates that the greater cell-mediated response using NPs made for a better adjuvant, however, cell dependence was not shown beyond cytokine analysis.

Response 3:

We sincerely thank the reviewer for their insightful comment regarding the cell-mediated immune response in our study. We agree that demonstrating the cell dependence of the observed immune response beyond cytokine analysis would strengthen our findings. However, our current study was primarily designed to compare the protective efficacy of two different antigens (rHlaH35L and rSpam) formulated with PLGA-PEG 25% NPs as an adjuvant, using alum as a positive control. While we observed a stronger cell-mediated immune response (as indicated by cytokine profiling) in the NP-adjuvanted groups compared to the alum-adjuvanted groups, we acknowledge that further mechanistic studies are needed to fully establish the cell dependence of this response.

Comments 4: Methods: What metrics are you using for physical condition (put in methods section). Also, I would say that mice were euthanized, rather than “executed”.

Response 4:

Thank you for raising these constructive questions and we appreciate it very much. The activity level; appetite; coat condition and skin integrity of immunized mice were used as physical condition metrics to evaluate the biocompatibility of nano-vaccines. These metrics have been added in revised manuscript. Please see line 204-205 in the revised manuscript. The corresponding results have been supplemented. Please see line 312-314.

Besides, the faulty expression of “executed”has been substituted as “euthanized”. Please see line 206; 242;266 in the revised manuscript.

Comments 5: What is “regular vaccination period was adopted” based on?

Response 5:

Thank you for raising this constructive question and we have replaced this imprecise expression with “In our research, mice were first inoculated at day 0, then boosted at day 14 and day 28.” in the revised manuscript. Please see line 318-319 in the revised manuscript.

We employed an immunization protocol with booster doses administered at two-week intervals based on following considerations. From a theoretical perspective, conducting the first booster immunization 14 days after the primary immunization allows for optimal utilization of memory cells generated following the initial exposure. This approach facilitates the further expansion of antigen-specific B cells and T cells and promotes antibody class switching (from IgM to IgG). The 14-day interval provides sufficient time for B cells to undergo affinity maturation, while multiple booster immunizations (e.g., on days 14 and 28) progressively enhance the affinity and specificity of the antibodies. If the interval between booster immunizations is too short (e.g., less than 7 days), the immune system may not fully recover, potentially leading to immune tolerance. Conversely, if the interval is too long (e.g., exceeding 4 weeks), the memory cells induced by the primary immunization may gradually decline, resulting in a weakened immune response. The 14-day interval represents a balanced time point that avoids immune tolerance while ensuring a robust immune response to the booster immunization. In experimental immunology, the 14-day interval has been extensively validated as an effective time point for eliciting strong immune responses across various antigens and animal models.

Studies on this immunization procedure and its related mechanisms can be found in the following reports:

  1. Thoryk, E.A.; Swaminathan, G.; Meschino, S.; Cox, K.S.; Gindy, M.; Casimiro, D.R.; Bett, A.J. Co-Administration of Lipid Nanoparticles and Sub-Unit Vaccine Antigens Is Required for Increase in Antigen-Specific Immune Responses in Mice. Vaccines 2016, 4, 47. https://doi.org/10.3390/vaccines4040047
  2. Kirsteina, A.; Akopjana, I.; Bogans, J.; Lieknina, I.; Jansons, J.; Skrastina, D.; Kazaka, T.; Tars, K.; Isakova-Sivak, I.; Mezhenskaya, D.; et al. Construction and Immunogenicity of a Novel Multivalent Vaccine Prototype Based on Conserved Influenza Virus Antigens. Vaccines 2020, 8, 197. https://doi.org/10.3390/vaccines8020197
  3. Wan, Y.; Zhang, Y.; Wang, G.; Mwangi, P.M.; Cai, H.; Li, R. Recombinant KRAS G12D Protein Vaccines Elicit Significant Anti-Tumor Effects in Mouse CT26 Tumor Models. Frontiers in oncology 2020, 10, 1326, doi:10.3389/fonc.2020.01326.
  4. Ghaemi, A.; Roshani Asl, P.; Zargaran, H.; Ahmadi, D.; Hashimi, A.A.; Abdolalipour, E.; Bathaeian, S.; Miri, S.M. Recombinant COVID-19 vaccine based on recombinant RBD/Nucleoprotein and saponin adjuvant induces long-lasting neutralizing antibodies and cellular immunity. Front Immunol 2022, 13, 974364, doi:10.3389/fimmu.2022.974364.
  5. Janeway, C. A., Travers, P., Walport, M., & Shlomchik, M. J. (2001). Immunobiology: The Immune System in Health and Disease. 5th edition. Garland Science.

Comments 6: What is an “ELISPOT histogram of IL-4 secretion”?

Response 6:

Thank you for raising this helpful question and we are sorry for this confusion we have made. The ELISPOT histograms were generated through quantitative analysis of the results shown in the Supporting Information (Figure S3.). The statistical calculations and graphical representations were performed to ensure the accuracy and reproducibility of the data visualization. This methodological approach allows for clear representation of the immune response observed in our ELISPOT assays while maintaining direct reference to the raw experimental data provided in the SI.

In response to the reviewer’ question, the puzzling expression “similar trend was found in the ELISPOT histogram of IL-4 secretion” has been replaced by “the secretion levels of IL-4 in the spleen of immunized mice exhibit a similar trend.” Please see line 355-357 in the revised manuscript.

Comments 7: Figure 4 is fuzzy, with small print that is difficult to read.

Response 7:

We sincerely thank the reviewer for pointing out the issue with Figure 4. We apologize for any inconvenience this may have caused. To address this concern, we have improved the quality and readability of Figure 4.

The revised version of Figure 4 is now included in the revised manuscript and below. We hope that the revised figure meets the reviewer’s expectations and provides a clearer representation of the data.

Figure 4. Immune activation response elicited by vaccines (a) Diagram of the immunization procedure and the sampling time points for immunoassay. (b) Antigen-specific serum IgG titer at day 17(n=6). (c) Antigen-specific serum IgG titer at day 35(n=15). (d) IL-4 secretion level revealed by ELISPOT at day 35(n=3). (e) IFN-γ secretion level revealed by ELISPOT at day 35(n=3). (f) IL-17A secretion level revealed by ELISPOT at day 35(n=3). (g) The percentage of lytic bacteria of serum antibody at day 35. All data were expressed as mean ± S.D. Statistical significance is indicated as p < 0.05 (* p < 0.05; *** p < 0.001; **** p < 0.0001).

Comments 8: Figure 5. Why are alum and NPs not included in the same studies?

Response 8:

We sincerely thank the reviewer for their question regarding Figure 5. We concur with the reviewer's insightful suggestion. We believe that the inclusion of both NPs and Alum in the same survival analysis would enhance the elucidation of the scientific questions. We have meticulously reanalyzed the experimental data and subsequently revised the corresponding figures to provide a more comprehensive and scientifically rigorous presentation of our findings.

Figure 5. Survival analysis after lethal challenge with S. aureus strain ATCC 25923 at a concentration of 2.56×108CFU per mouse (a) Diagram of immunization, lethal challenge and evaluation of protective efficacy. (b) Survival analysis between mice immunized with Alum-rHlaH35L and mice immunized with 25%NPs-rHlaH35L (n=10). (c) Survival analysis between mice immunized with Alum-rSpam and mice immunized with 25%NPs-rSpam (n=10). (d) Survival analysis between mice immunized with Alum-rHlaH35L+ Alum-rSpam and mice immunized with 25%NPs-rHlaH35L+ 25%NPs-rSpam (n=10). Survival rates were analyzed with Log-rank (Mantel–Cox) analysis. Statistical significance is indicated as p < 0.05 (* p < 0.05; *** p < 0.001).

Comments 9: Figure 6 & 7. Why are different antigens compared in different studies rather than the same study?

Response 9:

We sincerely appreciate the reviewer's insightful comment and we fully agree with the reviewer's valuable suggestion regarding the inclusion of different antigens within the same study. In response to this comment, we have systematically reanalyzed our data and reconstructed the corresponding figures to incorporate a comparative analysis of multiple antigens.

Figure 6. Survival analysis after lethal challenge with S. aureus strain ATCC 25923 at different bacteria concentration. (a) Diagram of immunization, lethal challenge and evaluation of protective efficacy. (b) Survival analysis among mice immunized with 25%NPs-rHlaH35L; 25%NPs-rSpam and 25%NPs-rHlaH35L+25%NPs-rSpam when mice were challenged with S. aureus strain ATCC 25923 at a concentration of 5.12×108CFU per mouse (n=10). (c) Survival analysis among mice immunized with 25%NPs-rHlaH35L; 25%NPs-rSpam and 25%NPs-rHlaH35L+25%NPs-rSpam when mice were challenged with S. aureus strain ATCC 25923 at a concentration of 1.024×109CFU per mouse (n=10). Survival rates were analyzed with Log-rank (Mantel–Cox) analysis. Statistical significance is indicated as p < 0.05 (**** p < 0.0001)

Comments 10: Fig. S2a what cell type was viability tested in? What assay was used?

Response 10: Thank you for raising this question. The cytotoxicity of the 25%NPs-Antigen was investigated by cell counting kit-8 (CCK-8) assay using fibroblast L929 cell line. L929 cells are sensitive to a variety of toxic substances and drugs, making them ideal for assessing compound cytotoxicity and biocompatibility.

Comments 11: Grammar needs to be improved. Also, parts of the manuscript were unclear based on word choices.

Response 11: We sincerely thank the reviewer for their careful reading and constructive feedback. We acknowledge that the grammar and clarity of certain parts of the manuscript needed improvement. We have now thoroughly revised the entire document to address these issues.

Reviewer 3 Report

Comments and Suggestions for Authors

The manuscript entitled: “Antigen-dependent adjuvanticity of PLGA-PEG 25% NPs for 2 enhanced vaccine efficacy” deals with a very interesting topic highlighting the adjuvant effect of PLGA-PEG nanoformulations conjugated with S. aureus antigens.  The manuscript is well written and includes appropriate experiments. However, there are some major and minor concerns that need to be addressed before suggesting its publication.

Minor:

·      Species names should be in italics.

·      Apart from antigen dose, the authors should also include information regarding the dose of 25%PLGA-PEG NPs that were administered to mice.

·      The grammar and syntax of the text should be improved. E.g. in lines 219-220 “the vaccine was immunized” should be replaced with “mice were immunized” and lines307-308: “The antibody was IgM dominated” should be replaced with “IgM was the dominant antibody subtype”.

·      Spaces between words or before citation number should be added where appropriate.

·      In paragraph 3.3 IgG levels between 25%-rSpam and rSpam groups are not significantly different. Please comment on that finding.

·      Please explain the finding of lines 409-411.

·      Line 463: body protection should be replaced with protection against challenge infection.

Major:

The discussion needs extensive revision. At first, the superiority of 25% PLGA-PEG as adjuvant compared to Alum should be discussed. Moreover, the difference between the exotoxin and cell-wall anchored antigen should be explained. The authors repeat their results attributing the higher survival rate of 25%NPs-rHlaH35L immunized mice compared to those that received 25%NPs-rSpam to higher immune response. However, these differences are not statistically significant. The authors should support this assumption with appropriate citations. In general, the discussion section should be enriched and the authors should compare their findings with similar papers with the same or other bacteria species.

Comments on the Quality of English Language

The quality of English language can be improved, ie grammar ant syntax. Please see comments for authors

Author Response

Dear reviewer,

We are very grateful for your constructive comments and suggestions for our manuscript entitled“Antigen dependent adjuvanticity of PLGA-PEG 25%NPs for enhanced vaccine efficacy.”(ID:3454176). Your comments are very valuable and helpful for improving our manuscript. In the following, the response to all the comments have been provided one by one.

We have tried our best to make all the revision clear and we hope that the revised manuscript can satisfy the requirements for publication.

Sincerely,

Li Fan; ChaoJun Song

Comments 1: The manuscript entitled: “Antigen-dependent adjuvanticity of PLGA-PEG 25% NPs for 2 enhanced vaccine efficacy” deals with a very interesting topic highlighting the adjuvant effect of PLGA-PEG nano formulations conjugated with S. aureus antigens.  The manuscript is well written and includes appropriate experiments. However, there are some major and minor concerns that need to be addressed before suggesting its publication.

Response 1:

Thank you for your positive feedback and constructive comments on our manuscript. We appreciate your recognition of the interesting topic and appropriate experimental design. Regarding your major and minor concerns, we have carefully listed and analyzed all the issues raised. We will clearly mark all changes in the revised manuscript and provide a point-by-point response in our response letter. Thank you again for your valuable time and expertise. We look forward to addressing your concerns and submitting an improved version of our manuscript.

Comments 2: Species names should be in italics.

Response 2: We sincerely thank the reviewer for their careful reading and valuable feedback. We have now revised the entire document to ensure that all species names are properly italicized in accordance with scientific writing conventions.

Comments 3: Apart from antigen dose, the authors should also include information regarding the dose of 25%PLGA-PEG NPs that were administered to mice.

Response 3:

Thank you very much for your careful review and valuable comments on our manuscript. According to the average loading efficiency of two antigens and antigen dosage, we calculated the dosage of nanoparticle:

For single antigen groups (rHlaH35L or rSpam), the number of nanoparticles administered per mouse was:

25 μg ÷ 1.1% (rHlaH35L average loading efficiency) = 2272.7 μg NPs

25 μg ÷ 1.8% (rSpam conjugation efficiency) = 1388.9 μg NPs

For the combined antigen group (rHlaH35L + rSpam), the total amount of nanoparticles administered per mouse was:

(25 μg ÷ 1.1%) + (25 μg ÷ 1.8%) = 2272.7 μg + 1388.9 μg = 3661.6 μg NPs

We have added this nanoparticle dosage information to the Materials section. Please see line 222-226 in the revised manuscript.

Comments 4: The grammar and syntax of the text should be improved. e.g. in lines 219-220 “the vaccine was immunized” should be replaced with “mice were immunized” and lines307-308: “The antibody was IgM dominated” should be replaced with “IgM was the dominant antibody subtype”.

Response 4: Thank you for your constructive question which helps us to improve the quality of the manuscript. The impropriate expressions have been corrected one by one (Please see line 322 in the manuscript.). Besides, the authors checked the manuscript thoroughly to improve the grammar and syntax.  

Comments 5: Spaces between words or before citation number should be added where appropriate.

Response 5: We sincerely thank the reviewer for their careful reading and valuable feedback. We have now thoroughly revised the entire document to ensure that spaces are added where appropriate

Comments 6: In paragraph 3.3 IgG levels between 25%-rSpam and rSpam groups are not significantly different. Please comment on that finding.

Response 6: We sincerely appreciate the reviewer’s attention to the comparison of IgG levels between the 25%-rSpam and rSpam groups. We would like to clarify that both on day 17 and day 35 after the first immunization there is indeed a significant difference in IgG levels between these two groups (p < 0.05, as indicated in Figure 4b, 4c). Please see line 331-334 and line 350-351 in the revised manuscript.

Comments 7: Please explain the finding of lines 409-411.

Response 7:

Thank you for raising this valuable question. we propose the following potential explanations: Based on the antibody titer and bacteriolysis assay results, the neutralizing antibody titers in the serum of the combined antigen group (25%NPs-rHlaH35L+25%NPs-rSpam) were slightly higher than those in the 25%NPs-rHlaH35L group. These neutralizing antibodies may have induced bacterial lysis through complement-mediated antibody-dependent cellular cytotoxicity. The release of toxic substances (e.g., SEB) from lysed bacteria could impose an additional physiological burden on the mice, indirectly leading to the decreased survival rate in the combined antigen group. Besides, rSpam might have obscured the critical protective epitopes of rHlaH35L through steric hindrance or immune modulation mechanisms, or it could have induced a form of immune suppression, thereby attenuating the vaccine's protective efficacy.

The precise mechanisms require further experimental validation. We will take this question into consideration seriously and plan to investigate these possibilities in future studies to clarify the potential impact of rSpam on the protective efficacy of the vaccine.

Comments 8: Line 463: body protection should be replaced with protection against challenge infection.

Response 8: Thank you for your helpful question. We have revised the Discussion Section and we have checked the revised manuscript thoroughly to avoid improper expression.

Comments 9: The discussion needs extensive revision. At first, the superiority of 25% PLGA-PEG as adjuvant compared to Alum should be discussed. Moreover, the difference between the exotoxin and cell-wall anchored antigen should be explained. The authors repeat their results attributing the higher survival rate of 25%NPs-rHlaH35L immunized mice compared to those that received 25%NPs-rSpam to higher immune response. However, these differences are not statistically significant. The authors should support this assumption with appropriate citations. In general, the discussion section should be enriched and the authors should compare their findings with similar papers with the same or other bacteria species.

Response 9: We deeply thank the reviewer for their detailed and constructive feedback. We agree that the Discussion section needed extensive revision to better highlight the significance of our findings and provide a more comprehensive comparison with the literature. We have thoroughly revised the Discussion section to address the reviewer's comments

Comments 10: The quality of English language can be improved, ie grammar and syntax. Please see comments for authors

Response 10:We sincerely thank the reviewer for their careful reading and constructive feedback. We acknowledge that the grammar and clarity of certain parts of the manuscript needed improvement. We have now thoroughly revised the entire document to address these issues.

Round 2

Reviewer 3 Report

Comments and Suggestions for Authors

The authors responded to my comments at an adequate level and therefore I recommend their manuscript for publication.

Comments on the Quality of English Language

The use of English language was improved

Author Response

Dear reviewer,

Thank you very much for your positive feedback and for recommending our manuscript for publication. We greatly appreciate the time and effort you have dedicated to reviewing our work and providing valuable comments, which have significantly improved the quality of our manuscript.

Best regards,

Li Fan